# Current Distribution of the Invasive Kelp *Undaria pinnatifida* (Harvey) Suringar, 1873 Along Artificial and Natural Habitats in North Portugal—Impacts and Mitigation Initiatives

**DOI:** 10.3390/plants14050658

**Published:** 2025-02-21

**Authors:** Marcos Rubal, Jesús Fernández-Gutiérrez, Diego Carreira-Flores, Pedro T. Gomes, Puri Veiga

**Affiliations:** 1Centre of Molecular and Environmental Biology (CBMA/ARNET), Department of Biology, University of Minho, 4710-057 Braga, Portugal; marcos.rubal@bio.uminho.pt (M.R.); jgutierrez@ciimar.up.pt (J.F.-G.); diego.carreira@bio.uminho.pt (D.C.-F.); pagomes@bio.uminho.pt (P.T.G.); 2CIIMAR/CIMAR LA, Interdisciplinary Centre of Marine and Environmental Research, University of Porto, Terminal de Cruzeiros do Porto de Leixões, 4450-208 Matosinhos, Portugal; 3Department of Biology, Faculty of Sciences, University of Porto, 4099-002 Porto, Portugal; 4Laboratorio Biodiversidade (GI-1934 TB), Instituto de Biodiversidade Agraria e Desenvolvemento Rural (IBADER), Campus Terra, Universidade de Santiago, 27002 Lugo, Spain

**Keywords:** Atlantic Ocean, natural and artificial habitats, biological invasion, kelp, *Undaria pinnatifida*

## Abstract

The kelp *Undaria pinnatifida* is considered one of the 100 worst invasive species in the world. The presence of *Undaria* in Portugal was reported more than 20 years ago, but there is not recent detailed information about its distribution and impacts. The aims of this study are to provide updated data about the distribution of *Undaria* in marinas and natural habitats along the north Portuguese coast, to afford insights about *Undaria* impacts on native macroalgae and to test the efficiency of mitigation actions based on *Undaria* removal. Mitigation measures were implemented in a marina where a native kelp was recently displaced by *Undaria*. The results showed stable populations of *Undaria* in marinas, but few ephemeral ones on natural rocky shores. The observed distribution pattern suggests an important role of salinity and wave exposure in shaping the distribution of *Undaria*. Moreover, *Undaria* was able to displace a native kelp and overgrow mussels in marinas, while on natural rocky shores, it was able to overgrow *Gongolaria baccata* but not native kelps. Finally, mitigation actions resulted in a partial recolonization by the native kelp.

## 1. Introduction

The Asian kelp *Undaria pinnatifida* (Harvey) Suringar, 1873 (hereafter, *Undaria*) is a native species from east China, Japan, Korea and southeast Russia [1,2,3]. Due to its economic value, *Undaria*, also known commercially as “Wakame”, was intentionally introduced at different points on the Atlantic shores of France for commercial exploitation [4]. Although the commercial production of *Undaria* failed, wild populations of this species were found in areas adjacent to the experimental culture sites [5,6]. However, this was not the only introduction instance of *Undaria* in Europe; previously, it was accidentally introduced onto the Mediterranean French coast, probably due to the importation of oysters [7]. Since these early events, *Undaria* has extended its range of distribution along the European Atlantic shores, northwards to the northern Wadden Sea [8] and southwards to central Portugal [9]. Out of the European region, during the last decade of the XX century and the first years of the current century, the presence of *Undaria* was also reported in the Australasian region (see review in [10]), the Patagonian region, southern Argentina [11], northwestern USA [12] and, more recently, Mexico [13].

Regarding the areas where *Undaria* has been introduced, there are a few studies about its effects on benthic native assemblages. Moreover, the available studies about the impact of *Undaria* on native assemblages showed contrasting results depending on the study area and experimental methodology. Most studies exploring *Undaria* impacts compared epifaunal assemblages harbored by native macroalgae and *Undaria*. Among these studies, Raffo et al. [14] found a lower diversity of epifauna on the holdfasts of *Undaria* than on the holdfasts of *Macrocystis pyrifera* (Linnaeus) C. Agardh, 1820 in Argentina but a similar diversity of epiphytic macroalgae. Similarly, Arnold et al. [15] in the UK found that *Undaria* harbored a lower diversity of invertebrates than native perennial kelps but similar diversity to that of an annual native kelp species. In New Zealand, Suárez-Jiménez et al. [16] reported similar epifaunal diversity in *Undaria* and native canopies with simple morphology. However, when compared with structurally complex native canopies, the diversity harbored by *Undaria* was lower [16]. Finally, Irigoyen et al. [17] performed an *Undaria* removal experiment in Argentina and found higher diversity and abundance of invertebrates in plots with *Undaria* than in the removal plots. However, this positive effect may have resulted from the lack of canopies in the removal plots, leading to a low habitat complexity.

Regarding the impacts of *Undaria* on native macroalgal assemblages, Casa et al. [18] found a dramatic decrease in native macroalgal diversity in areas of Argentina invaded by *Undaria*. However, Forrest and Taylor [19] reported little impact on invaded low shore assemblages in New Zealand, though the authors warned about limitations of the impact control design used to evaluate the *Undaria* effects on native assemblages [19]. Despite these contrasting results, *Undaria* is nowadays included in the list of the 100 of the world’s worst invasive species [20]; particularly in Europe, it has been considered one of the 10 worst invasive species [21]. Unfortunately, like many other non-indigenous marine species, the information about *Undaria* invasion is scarce and geographically biased, making it very difficult to develop management and mitigation measures to reduce its impact on native systems [22,23].

The first record of *Undaria* on the Atlantic shores of the Iberian Peninsula was in Galicia in 1988 [24], and later, it was found at different points of the north coast of Spain in 1995 [25]. After these first records, *Undaria* extended its distribution range, particularly on the northwest coast of Spain [26,27]. The first record of *Undaria* in Portugal was circa 2000, but its distribution was restricted to marina facilities in Aveiro and Póvoa do Varzim [28]. Báez et al. [29] proposed a macroevolutionary model that identified the northern coast of Portugal as a very favorable habitat for the establishment of *Undaria*. However, a later study by Veiga et al. [9] did not find any wild *Undaria* population along the north coast of Portugal; in contrast, these researchers found a wild population towards the south, at Buarcos, in central of Portugal. To our knowledge, the potential negative effects of *Undaria* on the native assemblages in marinas or natural habitats have not been explored in North Portugal.

The objectives of this study are the following: (i) to provide updated data about the distribution of *Undaria* in natural and artificial habitats along the north Portuguese coast; (ii) to afford insights about the first *Undaria* impacts on native macroalgae; (iii) to contribute with results obtained from the implemented mitigation actions, based on *Undaria* removal, to potentially reduce its impacts.

## 2. Results

### 2.1. Environmental Conditions and Associated Macroalgae

The values of salinity, temperature and dissolved oxygen in the six studied marinas are reported in Table A1. The results of ANOVA analyses did not show significant differences in temperature and dissolved oxygen between the marinas in estuaries and those out of estuaries (Table 1). However, significant differences were found in salinity, as the marinas out of estuaries showed significantly higher values than the marinas within estuaries (Table A1).

In general, the studied marinas supported a low diversity of macroalgae. In this study, the presence of four macroalga taxa different from *Undaria* were reported (Table A2). It is remarkable the presence of the non-indigenous *Grateloupia turuturu* Yamada, 1941 in marinas out of estuaries and the presence of the native canopy *Fucus aff spiralis* Linnaeus, 1753 in all the marinas within estuaries.

### 2.2. Distribution of Undaria pinnatifida

The presence of *Undaria* sporophytes was reported in three (Vm, Pv and Le) of the six studied marinas (see Figure 1). The lack of *Undaria* in the three brackish marina facilities contrasts with the presence of this species in all the marinas with marine salinity.

Individuals of *Undaria* were found on many different substrates such as boat hulls (Figure 2A), pontoons (Figure 2B), ropes (Figure 2C) or buoys (Figure 2D).

The results of the ANOVA analyses did not show significant differences in biomass or length of *Undaria* individuals among the three studied marinas (Table 2).

However, significant variability was found for biomass and length among the studied sites within each marina (Figure 3).

It is remarkable that many *Undaria* individuals collected from the marinas were overgrowing mussels fixed on pontoons or other structures (Figure 4A).

Regarding the distribution of *Undaria* in natural habitats, this kelp was found in three localities, two in Carreço, and one in Canto Marinho, during the summer of 2024 (Figure 4B). The three localities where individuals of *Undaria* were found are characterized by very sheltered conditions (Figure 1F,G). It is remarkable that several individuals of *Undaria* collected on natural rocky shores were found overgrowing native canopy species such as *Gongolaria baccata* (S.G. Gmelin, 1768) Molinari and Guiry, 2020 (Figure 4C).

The results of the ANOVA analyses did not show significant differences in the density or biomass of *Undaria* among the three studied localities (Table 3).

However, significant variability was detected for biomass and density among sites within each studied locality (Figure 5).

During sampling in December 2024, the lack of *Undaria* individuals was reported in Carreço and Canto Marinho, where previously *Undaria* had occurred.

Finally, significant differences in *Undaria* length were found between populations from marinas and natural rocky shores (KS = 2.88; *p* < 0.0001). On one hand, the populations from the marinas showed a higher frequency of lengths between 0 and 60 cm than the natural populations (Figure 6). On the other hand, the frequency of longer lengths was greater in the natural populations (Figure 6).

### 2.3. Mitigation Experiment

During the first experimental removal, in June 2023, a total of 21.8 kg of *Undaria* was removed, with a medium individual length of 56.5 ± 19.5 cm (Figure 7). No individuals of *Saccharina latissima* (Linnaeus) C.E. Lane, C. Mayes, Druehl & G.W. Saunders, 2006 were found in this first removal. Therefore, the total elimination of *S. latissima* from the area can be considered the first negative impact of *Undaria* on native macroalgal species in Portugal. During the second experimental removal (i.e., September 2023), the amount of removed *Undaria* (7 kg) and its medium length (34.2 ± 15.4 cm) decreased (Figure 7). Moreover, two individuals of *S. latissima* about 40 cm in length were found on the floating pontoon of the marina. During the next two experimental removals (i.e., December 2023 and March 2024), no individuals of *Undaria* were found. However, two senescent individuals of *S. latissima*, different from those observed in September, were found in December. Additionally, one individual of *S. latissima* of 37 cm was found in March. Finally, in June 2024, 0.17 kg of *Undaria* with a medium length of 36.6 ± 10.04 cm was removed. On this last date, two individuals of *S. latissima,* about 22 and 35 cm long, were found.

## 3. Discussion

Founding populations of *Undaria* in new areas are frequently found in marinas, aquaculture facilities or other artificial structures [30,31]. This pattern was also found in North Portugal, where more than 20 years ago, the first record of *Undaria* was reported from a marina in Pv [28]. The results of this study confirmed the presence of *Undaria* in Pv and in two additional marinas (i.e., Vm and Le). The results showed that *Undaria* is well stablished in marinas in the north of Portugal, with no differences (biomass and length) among the populations in the three marinas examined. However, *Undaria* was consistently absent in marinas located in brackish water estuaries (i.e., Vb, Es and Af). The absence of *Undaria* in brackish marinas contrasts with the results of many experimental studies that pointed out that *Undaria* is tolerant to low salinity. Bollen et al. [32] found that *Undaria* showed broader tolerance to different combinations of temperature and salinity than native New Zealand kelps. Similarly, Peteiro and Sánchez [33] found that germlings and gametophytes of *Undaria* were tolerant to the short-term exposure to salinity values of about 11 and 6 psu. The explored brackish marinas were interspersed among those with marine salinity and sometimes a few kilometers apart (e.g., Vm and Vb). However, the brackish marinas were inside estuaries, and the estuarine circulation may have acted as a barrier to the arrival of *Undaria* propagules in these marinas. Another potential explanation for the lack of *Undaria* in the brackish marinas may be the presence of native canopies of the genus *Fucus* (Table A2) that may over-compete *Undaria* under brackish conditions.

Early populations of *Undaria* established on marinas or other artificial structures showed some ability to spread out from these artificial structures and invade natural habitats on exposed rocky shores [22,23,30]. For more than two decades since the first record of *Undaria* in North Portugal, no population of this species was found on natural rocky shores [9]; similar situations were found in other areas where *Undaria* did not manage to colonize natural rocky habitats outside marinas for years [31]. In the present study, we reported the first record of *Undaria* on two natural rocky shores in North Portugal. The presence of *Undaria* was restricted to three sheltered areas (Figure 1F,G); however, in tidepools a few meters apart exposed to the wave action, *Undaria* was absent. The limited distribution in very sheltered areas and the posterior disappearance of these populations after the first winter storms suggest that wave exposure is an important driver shaping the distribution of *Undaria* sporophytes on natural rocky shores. Similarly, Epstein and Smale [23] found that wave action was an important determinant of the presence of *Undaria* on natural rocky shores in the UK, with higher *Undaria* abundance in sheltered areas. In areas where *Undaria* was able to colonize wave-exposed rocky shores from marinas [30], the wave height showed an annual average value of 1.07 m, while in our studied area, the common wave height is between 1.5 and 2 m, and wave heights of 7 m are frequent during winter.

Due to the different characteristics of the marinas and natural rocky shores where *Undaria* was found, we used different sampling methodologies that limited the comparison of *Undaria*’s traits, such as biomass or density. However, we could compare the size structure of the populations in marinas and on natural rocky shores. While no significant differences were found in density, biomass or length among the populations in marinas or on rocky shores, *Undaria* in these two habitats showed a significantly different length structure. This difference agrees with [34], which also found morphological differences, including in blade length and thallus length, between populations exposed to different water flow velocity. Moreover, our study detected significant variability between sites within the examined marinas or natural rocky shores, which may be the result of adaptation to specific local environmental and/or biological conditions.

The competitive ability of *Undaria* in marinas is poorly studied, but Ref. [31] found that *Undaria* was able to over-compete native kelps and the Ascidian *Styela clava* Herdman, 1881. In our study, we recorded that *Undaria* eliminated a stable population of the native kelp *S. latissima*. Our observational data cannot assure that the arrival of *Undaria* was due to the previous local extinction of *S. latissima*; however, the recolonization of the marina by *S. latissima* after *Undaria* removal suggests that *S. latissima* was not locally extinct but only excluded from the marina by *Undaria*. Moreover, we found many *Undaria* individuals overgrowing mussels, similar to the results reported in [31], which found many individuals of *Undaria* overgrowing *S. clava*, which disappeared from those studied marinas after about six years. The competitive ability of *Undaria* on natural rocky shores has received much more attention, but contrasting results were found. A study performed in New Zealand by Morelissen et al. [35] found no significant effects of disturbance (i.e., native macroalgae removal) in the recruitment of *Undaria*, which colonized all examined sites similarly, independently of the treatment. However, South and Thomsen [36] in New Zealand and De Leij et al. [37] in the UK found that native canopy removal resulted in a significantly increased colonization by *Undaria*; in areas where the canopies were not removed, *Undaria* was also present but in low densities. In our field observations, we found that *Undaria* was present mixed with native kelps such as *Laminaria ochroleuca* Bachelot de la Pylaie, 1824 and *Saccorhiza polyschides* (Lightfoot) Batters, 1902 but never overgrew these two native kelp species. In contrast, several individuals of *Undaria* were found overgrowing the native *G. baccata* species that is a very abundant native canopy in low tidal pools in the north of Portugal [38], which provide a suitable habitat for *Undaria* invasion whenever sheltered conditions.

Almost all the eradication or control initiatives of *Undaria* have failed [39,40]. However, the objective of our *Undaria* removal was to facilitate the recolonization of the marinas by the native *S. latissima* species, and this objective was partially achieved. Despite a lower density than before *Undaria* invasion, *S. latissima* recolonized the area. Moreover, removing *Undaria* in artificial habitats will reduce the propagule pressure on surrounding natural rocky shores, making its establishment in natural habitats more difficult, as proposed by [23].

## 4. Materials and Methods

### 4.1. Studied Area

This study encompassed the north of Portugal from the Minho to the Douro estuaries (Figure 1A). This area is characterized by an almost straight coastline dominated by rocky shores, sandy beaches and mixed shores of rock and sandy patches. The coastline is interrupted by estuaries of different sizes, being the Lima, Cavado and Douro estuaries among those with the biggest size (Figure 1B). Due to its orientation, the coastline is very exposed to wave action, with the dominant swell directions being west and northwest, and a typical wave height between 1.5 and 2 m, reaching maximum values of about 7 m during winter storms. The north coast of Portugal shows a semidiurnal tidal regime, with the largest spring tides of about 3.5–4.0 m. The studied area is also subjected to a seasonal upwelling during the spring–summer period, which enhances primary production due to an increased nutrient supply [41].

### 4.2. Field and Laboratory Methods

To explore the current distribution of *Undaria* sporophytes on marina facilities, the main six marinas in the studied area were visited during June 2023 (Figure 1C–E). At each marina, six independent measures of salinity, temperature and dissolved oxygen were taken using a multi-sonde (HQ 40 d, Hach, Düsseldorf, Germany), about 30 cm below the water surface. Three marinas were located within estuaries under brackish water conditions: Viana-e (Ve), Esposende (Es) and Afurada (Af). The remaining three were located out of estuaries, under fully marine salinity: Viana-m (Vm), Póvoa de Varzim (Pv) and Leça da Palmeira (Le). At each of these marinas, the presence of *Undaria* was explored on available hard substrates such as pontoons, ropes, seawalls or vessels. Due to the great variability of substrates where *Undaria* was found (e.g., ropes, boats, buoys), a proper quantification of its density in the marinas was impossible. When it was present, four individuals from two different sites (tens of meters apart) were collected to explore differences in their biomass and length.

Moreover, the presence of *Undaria* in the natural habitats of intertidal rocky shores was studied. The rocky shores in the study area were visited monthly between May 2023 and May 2024. After May 2024, the rocky shores were visited every three months. The presence or absence of *Undaria* was recorded in every visit, and in the three localities where *Undaria* was present (Figure 1F,G), its abundance was estimated by counting the number of individuals in five quadrats of 1 m^2^ at two sites separated by tens of meters. All the individuals of *Undaria* within each quadrat were collected in labeled plastic bags and transported to the laboratory. The three localities where *Undaria* was found in July 2024 were revisited in December 2024 to explore the evolution of the *Undaria* populations.

The length of all the individuals collected from the marinas and the natural rocky shores was measured in the laboratory to the nearest cm. To calculate the biomass, individuals from both marinas and quadrats in the natural habitats were washed with tap water to remove salt residues and epifauna. After washing, they were dried at 70 °C for 48 h in an oven, and their dry weight (g) was determined in gr using a precision balance.

### 4.3. Mitigation Experiment

Mitigation measures were executed in the Vm marina. During the sampling in June 2023, the presence of *Undaria* was reported for the first time in this marina. This marina and adjacent areas were previously colonized by the native boreal kelp *S. latissima*. During sampling in June 2023, no individuals of *S. latissima* were observed in the marina or surrounding artificial hard bottoms. The north of Portugal is the southern boundary of the distribution of this native kelp, and the Iberian Peninsula populations of *S. latissima* show a high degree of isolation from the northern populations and, thus, have a unique genetic structure [42]. Moreover, a negative effect of global warming on these southern populations is expected, which can be exacerbated by the interactive effect of non-indigenous species like *Undaria*. For all these reasons, we decided to implement a mitigation action in this marina. This action consisted in the selective elimination of *Undaria* individuals to provide space and resources that potentially allowed *S. latissima* to recolonize this marina. The first experimental elimination was conducted in June 2023 and was repeated every three months till July 2024. On each date, the fresh biomass (kg) of all the *Undaria* individuals removed from the marina was estimated, and the length of at least 40 individuals (when available) was measured following the methodology previously described. To facilitate the recolonization of *S. latissima*, the individuals of this species were not removed and were only counted and measured in the field.

### 4.4. Statistical Analyses

Analysis of variance (ANOVA) was used to explore significant differences in the environmental parameters (i.e., salinity, temperature and dissolved oxygen) between marinas within estuaries and marinas out of estuaries. For these analyses, we considered habitat (estuarine vs, non-estuarine) as a fixed orthogonal factor and marina as a random factor nested in habitat, with three levels and 6 replicates. To explore differences in the biomass and length of *Undaria* among the studied marinas, a two-way model design was used, with marina as a fixed orthogonal factor with three levels (Vm, Pv and Le), and site as a random factor nested in marinas, with two levels. Four replicates were considered per site.

In order to explore significant differences in biomass and density of *Undaria* between the two natural rocky shores where *Undaria* was found, ANOVA was also used based on a two-way model design, with locality as a fixed and orthogonal factor with three levels (Ca1, Ca2 and Cm1) and site as a random factor nested in locality, with two levels. Five replicates per site were considered.

Before each ANOVA, Cochran’s C tests were conducted to check for homogeneity of variances. When possible, the data were transformed to remove the heterogeneity of variances.

To explore differences in the size structure of *Undaria* between marinas and natural rocky shores, their length frequency was compared using Kolmogorov–Smirnov tests (KS).

## 5. Conclusions

Marinas with marine salinity harbored stable and similar *Undaria* populations in terms of biomass and length. Instead, natural rocky shores only harbored ephemeral populations during summer, which disappeared in winter. *Undaria* in North Portugal was only present in sheltered areas like marinas or in very specific locations on rocky shores. This limited distribution suggests an important role of wave exposure in shaping *Undaria* distribution. *Undaria* is able to overgrow some native species like mussels in marinas and *G. baccata* on rocky shores but is not able to overgrow native kelp species on rocky shores. Finally, *Undaria* displaced *S. latissima* in marinas, but the simple removal of *Undaria* was enough to allow *S. latissima* to recolonize this marina. Therefore, our data improve the knowledge of the Iberian populations of *Undaria* by providing empirical data that will be useful for future field studies and for management issues. Further investigations will be required to evaluate the long-term evolution of this species and its effects on ecosystems.

## Figures and Tables

**Figure 1 plants-14-00658-f001:**
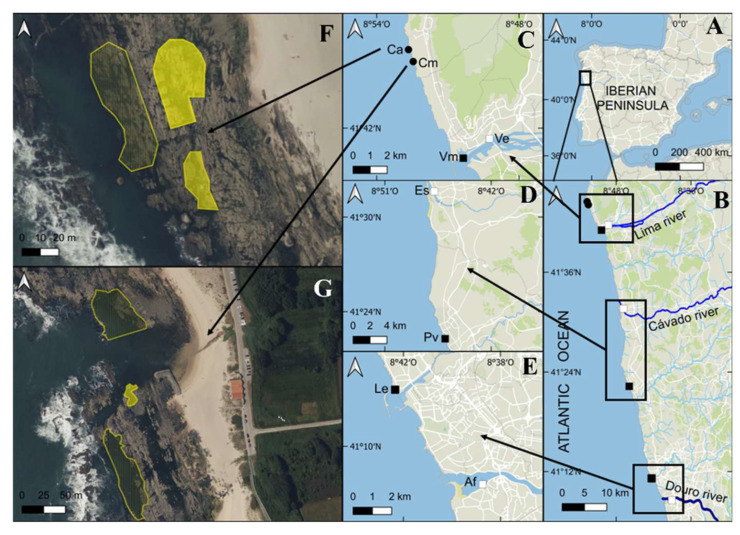
Studied area in North Portugal. (**A**) General view of the studied area. (**B**) Distribution of the studied marinas. The black squares are marinas with marine salinity, the white squares represent marinas with brackish salinity. (**C**) Location of the studied marinas on natural rocky shores (black circles); Carreço (Ca), Canto Marinho (Cm) and marinas of Viana-m (Vm) and Viana-e (Ve). (**D**) Location of the marinas Esposende, (Es) and Póvoa de Varzim (Pv). (**E**) Location of the marinas Leça da Palmeira (Le) and Afurada (Af). (**F**) Detail of the studied locations in Ca. (**G**) Detail of the studied locations in Cm. In (**F**,**G**), the yellow areas represent locations with *Undaria pinnatifida*; the empty yellow polygons represent locations without *Undaria pinnatifida*.

**Figure 2 plants-14-00658-f002:**
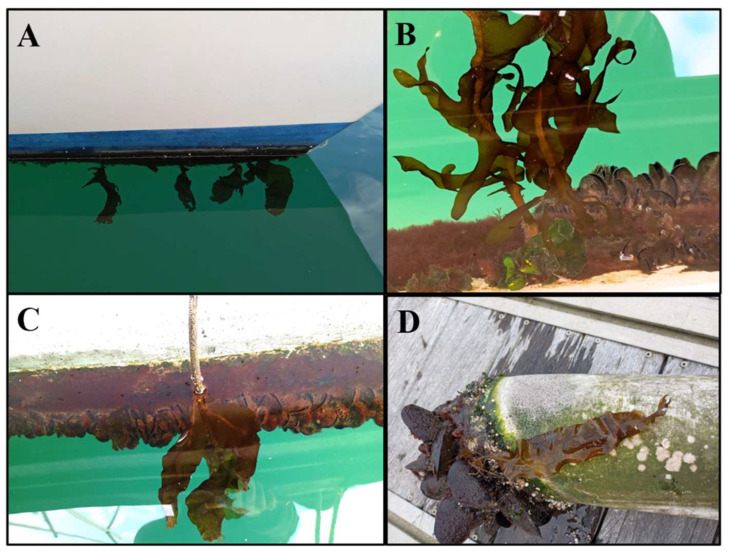
Different substrates where *Undaria pinnatifida* was found in this study. (**A**) Boat hull. (**B**) Pontoon. (**C**) Rope, (**D**) Buoy.

**Figure 3 plants-14-00658-f003:**
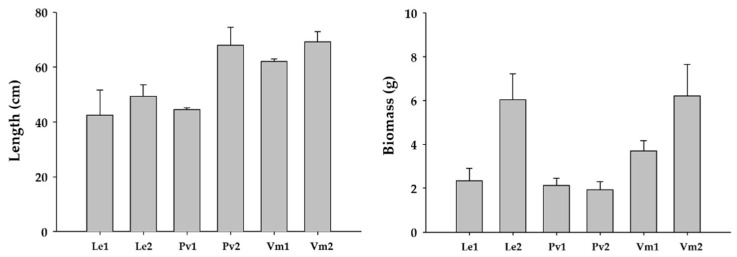
Mean values ± S.E. of the length and biomass of *Undaria pinnatifida* individuals found in the studied marinas.

**Figure 4 plants-14-00658-f004:**
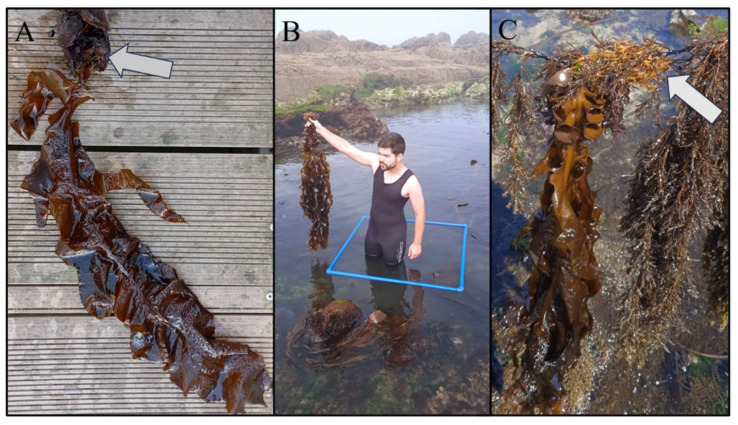
(**A**) *Undaria pinnatifida* overgrowing mussels in marinas; the white arrow shows mussels. (**B**) *Undaria pinnatifida* collected from natural rocky shores. (**C**) *Undaria pinnatifida* overgrowing native canopies; the white arrow shows the *Undaria* holdfast over the native canopy.

**Figure 5 plants-14-00658-f005:**
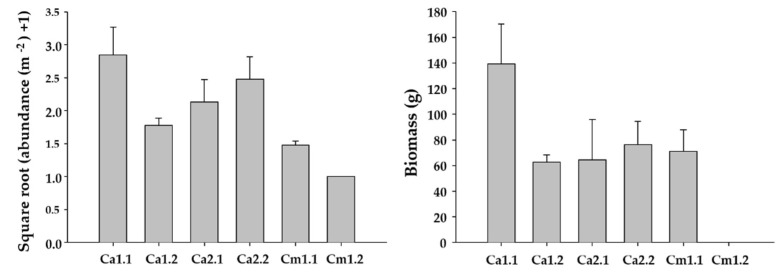
Mean values ± S.E. of the density and biomass of *Undaria pinnatifida* individuals found in the three studied natural localities. On the abscissa axis, the shores of Carreço (Ca) and Canto Marinho (Cm) are indicated. The first number indicates the locality, and the second number represents the site within each locality.

**Figure 6 plants-14-00658-f006:**
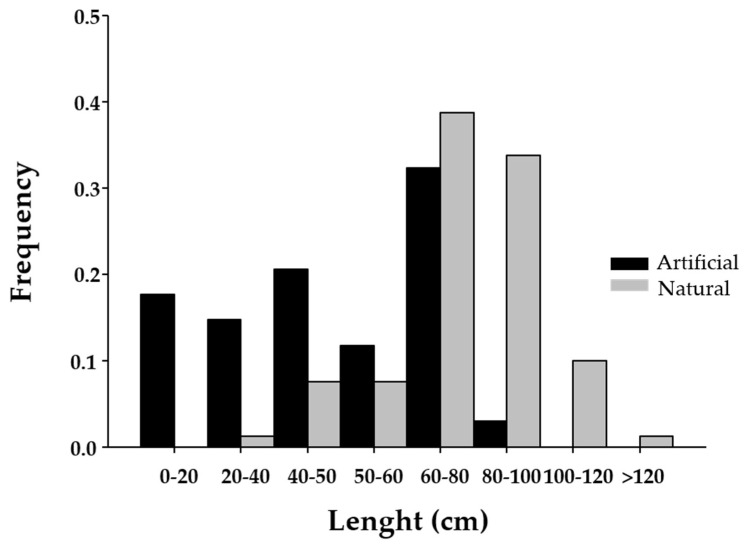
Length frequencies of *Undaria pinnatifida* individuals from marinas (grey) and natural rocky shores (black).

**Figure 7 plants-14-00658-f007:**
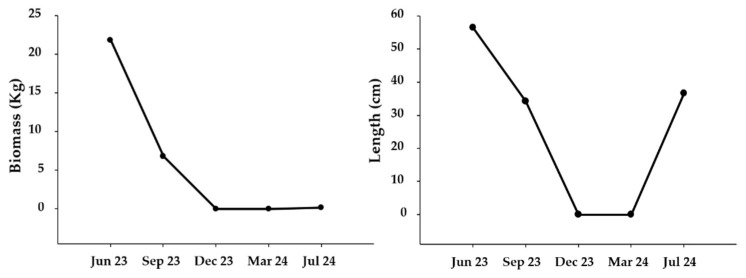
Temporal evolution of the biomass and length of the removed *Undaria pinnatifida* from the Vm marina.

**Table 1 plants-14-00658-t001:** Summary of the ANOVA analyses for environmental variables. Significant differences (*p* < 0.05) are indicated in bold.

Source	df	Salinity (psu)			Temperature (°C)			Dissolved Oxygen (mg L^−1^)			F-Versus
		MS	F	*p*	MS	F	*p*	MS	F	*p*	
Ha	1	3159.75	34.01	**0.004**	56.75	4.03	0.11	9.84	4.70	0.09	Ma (Ha)
Ma (Ha)	4	92.9	4.31	**0.007**	14.08	27.31	**0.0001**	2.09	12.93	**0.0002**	Residual
Residual	30	21.56			0.52			0.16			
Total	35										
Cochran’s Test		C = 0.41 (n.s.)			C = 0.32 (n.s.)			C = 0.41 (n.s.)			
Transformation		None			None			None			

**Table 2 plants-14-00658-t002:** Summary of the ANOVA analyses for the length and biomass of *Undaria pinnatifida* among the studied marinas. Significant differences (*p* < 0.05) are indicated in bold.

Source	df	Length (cm)			Biomass (g)			F-Versus
		MS	F	*p*	MS	F	*p*	
Ma	2	785.15	1.82	0.30	18.46	1.39	0.37	Si (Ha)
Si (Ma)	3	431.13	4.07	**0.02**	13.29	4.64	**0.01**	Residual
Residual	18	105.80			2.86			
Total	23				23			
Cochran’s Test		C = 0.52 (Not Significant)			C = 0.48 (Not Significant)			

**Table 3 plants-14-00658-t003:** Summary of the ANOVA analyses of the density and biomass of *Undaria pinnatifida* in the studied natural localities. Significant differences (*p* < 0.05) are indicated in bold.

Source	df	Density (Individuals m^−2^)			Biomass (g m^−2^)			F-Versus
		MS	F	*p*	MS	F	*p*	
Lo	2	3.81	3.05	0.19	10,708.61	1.16	0.42	Si (Ar)
Si (Lo)	3	1.25	3.55	**0.03**	9230.82	4.28	**0.01**	Residual
Residual	24	0.35			2155.40			
Total	29							
Cochran’s Test		C = 0.41 (Not Significant)			C = 0.38 (Not Significant)			
Transformation		Sqrt(X + 1)			None			

## Data Availability

The datasets presented in this article are not readily available because the data are part of an ongoing study. Requests to access the datasets should be directed to the corresponding author.

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
