# Peer review of "Current Distribution of the Invasive Kelp *Undaria pinnatifida* (Harvey) Suringar, 1873 Along Artificial and Natural Habitats in North Portugal—Impacts and Mitigation Initiatives"

_plants, 2025, doi:10.3390/plants14050658_

Round 1

Reviewer 1 Report

Comments and Suggestions for Authors

General comments

Some important references (Meretta et al, 2012; Dellatorre et al., 2014; Minchin, Nunn, 2014; Epstein, Smale, 2017-different from that you cite; James 2016), that could be useful for Discussion, were not considered. Moreover, it could be interesting to have data on environmental conditions (temperature, transparency, nutrient) of the studied sites, but also on the algal community associated to Undaria. It could be interesting to see differences or similarities among the sites. 

Specific comments

Title. I suggest to modify as “Current distribution of the invasive kelp…….”

Abstract

Line 15. I suggest to add here “(Harvey) Suringar, 1873 (hereafter Undaria)”

Line 17. I suggest to modify as “recent detailed information” at least about distribution (see Veiga et al., 2014)

Line 18. I suggest marinas instead of artificial

Line 20. I suggest to add “based on Undaria removal”

Lines 20-21. This sentence is redundant

Line 27. I suggest to eliminate here “based on Undaria removal” (see comment Line 20)

Keywords: I suggest to change as “Atlantic Ocean; Natural and artificial habitats; Biological invasion; Kelp; Undaria pinnatifida

Introduction

Lines 44-45. I suggest to add some useful references (Meretta et al, 2012; Dellatorre et al., 2014; Minchin, Nunn, 2014; Epstein, Smale, 2017; James 2016)

Lines 46-47. I suggest to modify this sentence, for instance as “In the areas where Undaria has been introduced, there are a few studies….” Here some useful references: Epstein, Smale, 2017 “Undaria pinnatifida: A case study to highlight challenges in marine invasion ecology and management”; James 2016 “A review of the impacts from invasion by the introduced kelp Undaria pinnatifida”; Minchin, Nunn, 2014 “The invasive brown alga Undaria pinnatifida (Harvey) Suringar, 1873 (Laminariales: Alariaceae), spreads northwards in Europe”

Lines 69-72. I suggest some useful references: the Review of Epstein, Smale, 2017 “Undaria pinnatifida: A case study to highlight challenges in marine invasion ecology and management”; James 2016 “A review of the impacts from invasion by the introduced kelp Undaria pinnatifida”

Line 87. I suggest to add “based on Undaria removal”

Results

Lines 91-92. Please, explain better this sentence

Figure 1, Lines 99-100. This sentence also applies to (G)

Line 126. Gongolaria baccata

Line 127, Line 129, Table 2. Density instead of abundance

Lines 136-137. Did the Authors look for gametophyte?

Line 141. I suggest “of longer lengths was greater”

Figure 6. Length instead of Size

Line 144. Length instead of Size

Line 149. See comment of Line 283

Lines 150-152. This is not a result but a comment. Moreover, do the Authors have any evidence of that?

Lines 155-156. Did the Authors look for gametophyte?

Discussion

Lines 191-194. It must also be considered the life cycle of Undaria, with a microscopic gametophyte which is not visible

Line 202, Line 204. Density instead of abundance

Lines 208-210. Are the sites different from environmental and biological point of view?

Lines 213-215. Do the Authors have any evidence of responsibility of Undaria? Is it possible that Undaria colonized a space left by S. latissima ? The opportunistic nature of Undaria seems to be unable to invade established kelp beds, but it colonizes rapidly when disturbance affects kelp canopies. Once established, dense stands of Undaria appear to inhibit kelp recruitment.

Line 230. G. baccata  

Line 233. The management of Undaria populations may be most effective by targeting the cause of Kelp disturbance.

Materials and Methods

Lines 253-274. The methodology used for marinas and rocky shores is different in term of sampling time and number of collected individuals, so it is not possible to compare the obtained data

Line 261. I suggest to eliminate this sentence, it is redundant

Line 262. How far were the two sites?

Line 263-265. I suggest to move it after Lines 259-261

Line 270. Density instead of abundance

Lines 270-273. Do the Authors estimate abundance and collect individuals in four quadrats of 1 m2 at two sites each sampling time?

Line 271. “in four quadrats of 1 m2 “ Are quadrats considered replicates?

Line 283. I suggest to change as “Saccharina latissima (Linnaeus) C.E.Lane, C.Mayes, Druehl & G.W.Saunders”. I think it is more correct here than in the Results (Line 149)

Lines 289-292. Do the Authors have any evidence about the interactive effects or about the role of Undaria in the absence of S. latissima?

Lines 293-296. I suggest to specify the area they removed. Did the Authors remove the same area or it was different each time?

Lines 307-308. The Authors consider here five replicates, see comment of Line 271. Why do the Authors use different replicates for marinas and rocky shores?

Conclusions

Lines 322-323. I think that further studies are needed for these conclusions

Comments on the Quality of English Language

English should be also checked

Author Response

1-Some important references (Meretta et al, 2012; Dellatorre et al., 2014; Minchin, Nunn, 2014; Epstein, Smale, 2017-different from that you cite; James 2016), that could be useful for Discussion, were not considered. Moreover, it could be interesting to have data on environmental conditions (temperature, transparency, nutrient) of the studied sites, but also on the algal community associated to Undaria. It could be interesting to see differences or similarities among the sites. 

Information about environmental conditions (salinity, temperature and dissolved oxygen) was added in the results and in the appendix A (Table A1). Macroalgae were scarce on marinas but the information about the presence of other macroalgae different to Undaria were included in the table A2 included in the appendix A.

Some of the references proposed by the referee were included (see response to next comments).

Specific comments

2-Title. I suggest to modify as “Current distribution of the invasive kelp…….”

Suggestion included.

Abstract

3-Line 15. I suggest to add here “(Harvey) Suringar, 1873 (hereafter Undaria)”

We think that the authority of the species and the definition of the abbreviate name should be included in the main text of article. The abstract can be understood without this information but many readers may read directly the article (overlooking the abstract) and they will miss this information.

4-Line 17. I suggest to modify as “recent detailed information” at least about distribution (see Veiga et al., 2014)

Correction done.

5-Line 18. I suggest marinas instead of artificial

Correction done.

6-Line 20. I suggest to add “based on Undaria removal”

Text added.

7-Lines 20-21. This sentence is redundant

Sentence was deleted.

8-Line 27. I suggest to eliminate here “based on Undaria removal” (see comment Line 20)

Text was removed.

9-Keywords: I suggest to change as “Atlantic Ocean; Natural and artificial habitats; Biological invasion; Kelp; Undaria pinnatifida

Suggestions were accepted and included in the manuscript.

Introduction

10-Lines 44-45. I suggest to add some useful references (Meretta et al, 2012; Dellatorre et al., 2014; Minchin, Nunn, 2014; Epstein, Smale, 2017; James 2016)

Thank you very much for the suggestion but in this point of the introduction we think that the included references are good enough to illustrate the distribution of Undaria and new references would be redundant.

11-Lines 46-47. I suggest to modify this sentence, for instance as “In the areas where Undaria has been introduced, there are a few studies….” Here some useful references: Epstein, Smale, 2017 “Undaria pinnatifida: A case study to highlight challenges in marine invasion ecology and management”; James 2016 “A review of the impacts from invasion by the introduced kelp Undaria pinnatifida”; Minchin, Nunn, 2014 “The invasive brown alga Undaria pinnatifida (Harvey) Suringar, 1873 (Laminariales: Alariaceae), spreads northwards in Europe”

We have modified the sentence following referee´s suggestions. We did not include any additional reference here because in the next lines we detailed the studies that explore the impacts of Undaria on different benthic assemblages.

12-Lines 69-72. I suggest some useful references: the Review of Epstein, Smale, 2017 “Undaria pinnatifida: A case study to highlight challenges in marine invasion ecology and management”; James 2016 “A review of the impacts from invasion by the introduced kelp Undaria pinnatifida”

References were included.

13-Line 87. I suggest to add “based on Undaria removal”

Correction done.

Results

14-Lines 91-92. Please, explain better this sentence

Sentence was modified.

15-Figure 1, Lines 99-100. This sentence also applies to (G)

This point was clarified.

16-Line 126. Gongolaria baccata

Name was corrected.

17-Line 127, Line 129, Table 2. Density instead of abundance

Correction was done.

18-Lines 136-137. Did the Authors look for gametophyte?

This work is only focused on the macroscopic sporophytes.

19-Line 141. I suggest “of longer lengths was greater”

Correction done.

20-Figure 6. Length instead of Size

Corrected.

21-Line 144. Length instead of Size

Correction done.

22-Line 149. See comment of Line 283

Response on comment of line 283

23-Lines 150-152. This is not a result but a comment. Moreover, do the Authors have any evidence of that?

This is a field observation. After removing Undaria we noticed the lack of S. latissima that was the dominant macroalga in the marina for many years before the arrival of Undaria (see also response to comment lines 213-215.

24-Lines 155-156. Did the Authors look for gametophyte?

See response to comment lines 136-137.

Discussion

25-Lines 191-194. It must also be considered the life cycle of Undaria, with a microscopic gametophyte which is not visible

We clarified that this sentence refers to Undaria sporophytes.

26-Line 202, Line 204. Density instead of abundance

Correction done.

27-Lines 208-210. Are the sites different from environmental and biological point of view?

Sites within each location were separated by meters and thus the environmental and biological conditions were similar.

28-Lines 213-215. Do the Authors have any evidence of responsibility of Undaria? Is it possible that Undaria colonized a space left by S. latissima ? The opportunistic nature of Undaria seems to be unable to invade established kelp beds, but it colonizes rapidly when disturbance affects kelp canopies. Once established, dense stands of Undaria appear to inhibit kelp recruitment.

We considered this possibility see lines 217-222. However, as we stated in that paragraph S. latissima only disappeared after the colonization of Undaria, not before. Moreover, when Undaria was removed S. latissima recolonize the marina proving that environmental conditions were still suitable for it and was not locally extinct. Similar observations were done by Farrell and Fletcher (see response to comente line233), showing that Undaria can over-compete native kelps and invertebrates in marinas without a previous disturbance in contrats to natural rocky shores.

29-Line 230. G. baccata  

Corrected.

30-Line 233. The management of Undaria populations may be most effective by targeting the cause of Kelp disturbance.

We agree that control of anthropogenic disturbances is useful for management of Undaria but, natural disturbances are impossible to control. In any case, disturbance is critical in exposed rocky shores, Undaria is able to eliminate native kelp from marinas without any previous disturbance see Farrell, P.; Fletcher, R.L. An investigation of dispersal of the introduced brown alga Undaria pinnatifida (Harvey) Suringar and its competition with some species on the man-made structures of Torquay Marina (Devon, UK). J. Exp. Mar. Biol. Ecol. 2006, 334, 236–243. https://doi.org/10.1016/j.jembe.2006.02.006

Materials and Methods

31-Lines 253-274. The methodology used for marinas and rocky shores is different in term of sampling time and number of collected individuals, so it is not possible to compare the obtained data

We only compared length frequencies by a Kolgomorov Smirnov test that can be used with different number of samples by treatment. We agree that the time difference is important but during June 2023 there was not available populations in natural rocky shores. In any case, both collections were done during summer avoiding seasonal changes that are bigger than inter annual ones.

32-Line 261. I suggest to eliminate this sentence, it is redundant

This sentence was removed.

33-Line 262. How far were the two sites?

Information was included.

34-Line 263-265. I suggest to move it after Lines 259-261

Change was made.

35-Line 270. Density instead of abundance

Change was done.

36-Lines 270-273. Do the Authors estimate abundance and collect individuals in four quadrats of 1 m2 at two sites each sampling time?

Yes.

37-Line 271. “in four quadrats of 1 m2 “ Are quadrats considered replicates?

Yes, but we sampled 5 not 4. Mistake was corrected.

38-Line 283. I suggest to change as “Saccharina latissima (Linnaeus) C.E.Lane, C.Mayes, Druehl & G.W.Saunders”. I think it is more correct here than in the Results (Line 149)

Considering the botanical code of nomenclature, we must include the full name of the species and the authorities the first time that the species is cited in the manuscript (i.e. line 149)

39-Lines 289-292. Do the Authors have any evidence about the interactive effects or about the role of Undaria in the absence of S. latissima?

We do not have any evidence of the interactive effects of global warming and Undaria and that why we state “it is expected” or “that can be exacerbated” For the role of Undaria in the absence of S. latissima see response to lines 213-215

40-Lines 293-296. I suggest to specify the area they removed. Did the Authors remove the same area or it was different each time?

We always removed the same area (i.e. the whole marina).

41-Lines 307-308. The Authors consider here five replicates, see comment of Line 271. Why do the Authors use different replicates for marinas and rocky shores?

The correct number of quadrats was 5. We had to adapt the sampling effort at each habitat in function of the abundance and size of Undaria. Both in marinas and rocky shores we sampled two sites by location. In rocky shores we can use quadrats but in marinas we had to collect individuals because we cannot use quadrats in ropes or other substrates. We had to adapt the sampling methods to each habitat.

Conclusions

42-Lines 322-323. I think that further studies are needed for these conclusions

We observed the recolonization of S. latissima after the Undaria removal. Of course, the long-term sustainability of new S. latissima populations should be explored by future research as we stated at end of the manuscript.

Reviewer 2 Report

Comments and Suggestions for Authors

Expanding distribution of invasive alien species in coastal areas is a problem in many areas. This study investigated the distribution of Undaria in the northern coastal zone of Portugal, as well as the effectiveness of its seaweed removal. I did not see any major problems with the significance or methodology of the study. However, I felt that the structure and writing of the paper needed major revisions.

Individual comments are as follows:

Line 17: It is described as “impacts”, but to what are the impacts intended?

Lines 18 and 20: Line 18 lists natural and artificial and Line 20 lists marinas and natural. It would be easier to understand if either order were switched.

Line 47: It is listed as “assemblages”, does it mean “seaweed assemblages”?

Lines 62-72: “However” appears three times in this paragraph. Please consider replacing it with another word.

Results section: Results are shown in the next section of Introduction, why not Methods?

The order of Introduction, Methods, Results, Discussion, and Conclusion is the easiest to understand.

Line 91: At the end of the sentence, it states “see Figure 1”, please make sure the grammar of this sentence is correct.

Line 99-100: You mention the yellow polygons that indicate the presence or absence of Undaria, how exactly were these polygons created? Did you visually trace the image?

Figure 5: Numbers such as 1.1 and 1.2 are shown. What does this mean?

Figure 5: Do you have a value for the Cm1.2 biomass on the right side of the figure on the right?

Figure 6: It would be easier to understand if the order of Natural and Artificial in the figure is reversed.

Line 153: Please confirm that “34.2 c 15.4 cm” is the correct notation.

Figure 7: Is it possible to show the standard deviation in this figure? Also, the intersection of the vertical and horizontal axes would be easier to understand with zero.

Line 173: It is listed as “low salinity”, can you indicate how much is the salinity threshold?

Section 4.2: In several places, n=6 is listed. What does this mean?

Section 4.2: What instrument was used to measure salinity? What is the depth of salinity measurement? Also, shouldn't the salinity values be listed in the results section to begin with?

Comments on the Quality of English Language

Check for minor typos.

Author Response

Expanding distribution of invasive alien species in coastal areas is a problem in many areas. This study investigated the distribution of Undaria in the northern coastal zone of Portugal, as well as the effectiveness of its seaweed removal. I did not see any major problems with the significance or methodology of the study. However, I felt that the structure and writing of the paper needed major revisions.

Individual comments are as follows:

1-Line 17: It is described as “impacts”, but to what are the impacts intended?

 This is the introduction and thus, we can not provide a detailed description of the impacts. Along the manuscript we related how Undaria eliminates a native kelp species (Saccharina latissimi) from a marina and is able to overgrowth mussels and native canopies.

2-Lines 18 and 20: Line 18 lists natural and artificial and Line 20 lists marinas and natural. It would be easier to understand if either order were switched.

Correction was done.

3-Line 47: It is listed as “assemblages”, does it mean “seaweed assemblages”?

Undaria can have negative impacts on different assemblages as epifauna (see lines 49-61) or in native macroalgae and/or benthic macrobenthos (detailed in lines 62 to 65). To be more specific, we modified the sentence as “benthic assemblages” but at this point of the manuscript we are doing a general consideration and more detailed information can be found in the next lines.

4-Lines 62-72: “However” appears three times in this paragraph. Please consider replacing it with another word.

Correction was done.

5-Results section: Results are shown in the next section of Introduction, why not Methods? The order of Introduction, Methods, Results, Discussion, and Conclusion is the easiest to understand.

We agree with the referee, but this is the section order imposed by the journal.

6-Line 91: At the end of the sentence, it states “see Figure 1”, please make sure the grammar of this sentence is correct.

The sentence was revised and modified.

7-Line 99-100: You mention the yellow polygons that indicate the presence or absence of Undaria, how exactly were these polygons created? Did you visually trace the image?

The polygons were created in the basis of the geographical coordinates of the pools where Undaria was present or absent. However, we did not include this information in the figure caption as we think that it is not critical to understand the figure.

8-Figure 5: Numbers such as 1.1 and 1.2 are shown. What does this mean?

The meaning of the numbers was explained in the figure caption.

9-Figure 5: Do you have a value for the Cm1.2 biomass on the right side of the figure on the right?

The value of Cm1.2 was included but, it is very small in comparison with other localities and sites.

10-Figure 6: It would be easier to understand if the order of Natural and Artificial in the figure is reversed.

Modification was done.

11-Line 153: Please confirm that “34.2 c 15.4 cm” is the correct notation.

This mistake was corrected.

12-Figure 7: Is it possible to show the standard deviation in this figure? Also, the intersection of the vertical and horizontal axes would be easier to understand with zero.

In this figure we provide total values and thus there is not standard deviation values associated. Moreover, if place the zero value on the intersection of vertical and horizontal axes the graph will overlap the x axis on several points and will be difficult to see.

13-Line 173: It is listed as “low salinity”, can you indicate how much is the salinity threshold?

Lower values of salinity considered in the study were detailed.

14-Section 4.2: In several places, n=6 is listed. What does this mean?

 “n” is the standard representation of the number of replicas used to calculate the mean values.

15-Section 4.2: What instrument was used to measure salinity? What is the depth of salinity measurement? Also, shouldn't the salinity values be listed in the results section to begin with?

Information about the salinity and other environmental variables was added, including the depth. Salinity and other environmental variables were included in an annexe table and were presented in the beginning of the result section.

Round 2

Reviewer 1 Report

Comments and Suggestions for Authors

No further comments

Reviewer 2 Report

Comments and Suggestions for Authors

I confirm that you have responded appropriately to my peer review comments.